# Implementing person-centred outcome measures (PCOMs) into routine palliative care: A protocol for a mixed-methods process evaluation of The RESOLVE PCOM Implementation Strategy

Andy Bradshaw  ,[1] Martina Santarelli,[1] Assem M Khamis,[1] Kathryn Sartain,[1,2] Miriam Johnson,[1] Jason Boland  ,[1] Mark Pearson,[1] Fliss E M Murtagh[1]

¹Wolfson Palliative Care Research Centre, Hull York Medical School, University of Hull, Hull, UK
²York and Scarborough Teaching Hospitals NHS Foundation Trust, York, UK

**Correspondence to**
Andy Bradshaw;
andrew.bradshaw@hyms.ac.uk

## ABSTRACT

**Introduction** Person-centred outcome measures improve quality of care and patient outcomes but are used inconsistently in palliative care practice. To address this implementation gap, we developed the 'RESOLVE Implementation Strategy'. This protocol describes a process evaluation to explore mechanisms through which this strategy does, or does not, support the implementation of outcome measures in routine palliative care practice.

**Methods and analysis** Multistrand, mixed-methods process evaluation. Strand one will collect routine outcomes data (palliative Phase of Illness, Integrated Palliative care Outcomes Scale, Australia-modified Karnofsky Performance Status) to map the changes in use of outcome measures over 12 months (July 2021–July 2022). Strand two will collect survey data over the same 12-month period to explore how professionals' understandings of, skills in using and ability to build organisational practices around, outcome measures change over time. Strand three will collect interview data to understand the mechanisms underpinning/affecting our implementation strategy. Thematic framework analysis and descriptive statistics will be used to analyse qualitative and quantitative data, respectively.

**Ethics and dissemination** For strand one, ethical approval has been obtained (Cambridge REC, REF: 20/EE/0188). For strands two and three, ethical approval has been obtained from Hull York Medical School ethics committee (2105). Tailored feedback of study findings will be provided to participating sites. Abstracts and papers will be submitted to national/international conferences and peer-reviewed journals. Lay and policy briefings and newsletters will be shared through patient and public involvement and project networks, plus via the project website.

## INTRODUCTION
### Person-centred outcome measures in palliative care

Palliative care aims to alleviate suffering due to progressive, life-limiting medical conditions and improve quality of life through the adoption of a holistic, person-centred and multidisciplinary approach that is delivered across multiple

---

**Strengths and limitations of this study**

► This is the first process evaluation of an implementation strategy designed to facilitate the integration of person-centred outcome measures into routine palliative care practice.

► This protocol follows the Medical Research Council's guidance on process evaluations by adopting a systematic and theoretically informed approach to study design and conduct.

► We will adopt a multistrand, mixed-methods approach in which we will integrate routine outcomes, survey and focused interview data to demonstrate the mechanisms through which our intervention strategy may or may not work across different settings of palliative care.

► Since we only focus on implementing a core set of outcome measures, the applicability of findings may be limited to other outcome measures, many of which have measure-specific challenges to implementation.

---

settings (in-patient, outpatient, hospital, and community).[1] It is projected that by 2060, the global need for palliative care will double (from 26 million to 48 million people).[2] It is important, therefore, that we are able to clearly demonstrate the quality of palliative care services by showing if and how they are able to improve a person's symptoms, and sustain or improve well-being and functional ability. To do this, focusing on outcomes is essential.

In a healthcare context, an outcome refers to 'the change in a patient's current and future health status that can be attributed to preceding healthcare'.[3] Within palliative care, measuring outcomes is important because they allow us to evaluate whether the care that is given to patients and their families makes a difference to their quality of life.[4] In capturing this information, person-centred outcome measures (PCOMs)

**BMJ**

are considered the 'gold standard' for outcome measurement in palliative care.[5 6] These are standard, validated questionnaires—usually filled in by patients themselves—that provide healthcare professionals with important information on a person's own perception of their health status and well-being.[7] Moreover, because many patients with life-limiting illnesses may have impaired cognition and/or be too unwell to fill out PCOMs on their own accord,[8] PCOMs may sometimes be completed using proxy-reported ratings (eg, by healthcare professionals and/or a patient's family member).

Current evidence demonstrates the value of PCOMs in palliative care in helping to identify (often unrecognised), continually monitor, and take direct clinical action to address a person's needs/symptoms; facilitate communication between patients and clinicians, and improve outcomes.[4 7 9] This is through using PCOMs at individual/patient (eg, for clinical assessment), team (eg, focusing patient reviews and workload planning), service (eg, service evaluation/development and making business cases) and population (eg, benchmarking) levels of practice.[10] For these reasons, the European Association for Palliative Care Task Force on Outcome Measurement recommended that PCOMs should be firmly embedded into routine clinical practice across all settings in which palliative care is delivered.[6]

Despite the evidence underpinning the value of using PCOMs, they are used inconsistently in clinical practice, if at all. Reasons for this include issues such as time constraints, lack of training and knowledge, tools being perceived as burdensome, negative attitudes towards outcome measures, inefficient electronic systems, top-down approaches to implementation and fear of added work.[5 11–13] In addressing these challenges and facilitating the implementation of PCOMs into everyday clinical practice, various studies have provided tables of recommendations to consider before and during implementation.[5 12 13] These have recommended focusing on individual, management and organisational issues during the preparation and roll-out stages of implementation. To date, however, there has been no attempt to develop these into, and subsequently evaluate, an implementation strategy that may be applied across all palliative care settings.

### The RESOLVE project

The RESOLVE project is a multisite implementation and research programme aimed at implementing PCOMs into routine clinical practice in 11 specialist palliative care services across Yorkshire, England. These services vary in size, funding source (eg, charitable/NHS), location (eg, urban/semi-rural/rural) and span hospice in-patient, outpatient/day therapy and home-based/community settings. As part of this study, we conducted an exploratory qualitative study in which semi-structured interviews were used to understand the processes and causal mechanisms that underpin the successful implementation of PCOMs into routine practice.[13] Building on insights from this study about implementation and our collaborative

work with sites, this study protocol describes a process evaluation of the complex intervention—named 'The RESOLVE PCOM Implementation Strategy'—aimed at facilitating the integration of PCOMs into routine practice. It is important to evaluate these types of interventions to understand what does (and does not work) and in what contexts.

Process evaluations are useful in understanding the ways in which complex interventions work because they provide an insight into the causal mechanisms and contextual factors at play during implementation.[14] Currently, however, no process evaluations have been conducted that explore the impact of implementation strategies aimed to promote the uptake and sustained use of PCOMs within palliative care contexts.

Therefore, we aim to conduct a process evaluation that explores the mechanisms through which The RESOLVE PCOM Implementation Strategy does, or does not, support the implementation of PCOMs in routine palliative care practice. We aim to achieve this by answering three research questions:
1. Does The RESOLVE PCOM Implementation Strategy enhance the uptake and routine use of PCOMs?
2. Does The RESOLVE PCOM Implementation Strategy enhance healthcare professionals' individual and collective understanding of, skills in using, and ability to build organisational practices around, PCOMs?
3. If The RESOLVE PCOM Implementation Strategy is effective, through which mechanisms are its intended effects achieved?

## Methods and analysis
### Study design

A multistrand, mixed-methods, longitudinal process evaluation informed by The Medical Research Council guidance.[15] To do this, we will adopt a systematic approach in which we provide clear descriptions of, and collect relevant data pertaining to: (i) our complex intervention; (ii) the implementation process; (iii) mechanisms; (iv) context and (v) outcomes/results (see table 1). This will be achieved through three strands of work. An overview these can be seen in table 2.

### Core PCOMs set

The three outcome measures that we aim to implement are a nationally recognised and validated core set of PCOMs for specialist palliative care.[16] These measures are based on what patients and their families prioritise as important in advanced illness and have been endorsed by NHS England and Public Health England. They include the palliative Phase of Illness,[17] Integrated Palliative care Outcomes Scale (IPOS)[18 19] and Australia-modified Karnofsky Performance Status (AKPS).[20] More information about each of these measures can be seen in table 3.

The implementation of these PCOMs is not simply about increased use, but the wrap-around work that is needed for the effective and appropriate use of these measures to improve care. This includes organisational and team recognition of

**Table 1** Key components of a process evaluation and how this study has considered these

| Key components of process evaluations | Medical Research Council guidance[15] | This study |
|---|---|---|
| Describing the complex intervention and clarifying causal assumptions | 'A clear description of the intended intervention, how it will be implemented, and how it is expected to work, will ideally have been developed before evaluation' | We describe The RESOLVE person-centred outcome measure (PCOM) implementation strategy and have drawn on Normalisation Process Theory to explain the causal assumptions through which we expect it to work. |
| **Implementation: what is implemented, and how?** | ► 'Implementation process (how delivery is achieved; training, resources etc.)'<br>► 'What is delivered (fidelity, dose, adaptations, reach)' | **We have outlined, in detail, the features of our complex interventions using the template for intervention description and replication (TIDieR) checklist (see online supplemental file 1). This outlines what each component is, why we are doing it, who is delivering, how much they are delivering, where and when it is being delivered, how it may be tailored/modified and how well it will be delivered (eg, fidelity).** |
| **Mechanisms of impacts** | ► 'Participant responses to and interactions with the intervention'<br>► 'Mediators'<br>► 'Unexpected pathways and consequences' | **In understanding mechanisms of impacts, we are collecting three types of data:**<br>► **Routinely collected person-centred outcome measures data.**<br>► **Survey data.**<br>► **Focused interview data.** |
| **Context** | ► 'Contextual factors that shape theories of how the intervention works'<br>► 'Contextual factors that affect (and may be affected by) implementation, intervention mechanisms and outcomes'<br>► 'Causal mechanisms present within the context which act to sustain the status quo or potentiate effects' | **Analysis of focused interview data which provide an insight into the contextual factors and mechanisms (eg, settings of care or organisational characteristics) that affect (and are affected by) the implementation process.** |
| Outcomes | Analysis of process data, and integration of process and outcome data | We will combine results/analysis from quantitative and qualitative analyses to produce numerous outputs that answer the research questions of this study. |

Bold areas represent key components of a process evaluation according to Medical Research Council guidance.

why using measures is important, using the right measures at the right time, ensuring follow-up assessments using the measures, getting all teams/team members on board, having efficient feedback systems in place, having all members of the team understanding the purpose of PCOMs and embedding PCOMs into the 'cultural fabric' of how teams and services operate.[13]

### Description and causal assumptions of The RESOLVE PCOM Implementation Strategy

#### Description of complex intervention

While all participating sites collect PCOMs in some form, their use is variable. For example, some use all three measures but not consistently across settings, others may use certain measures consistently in some settings but not others, while some are seeking to introduce new measures that they do not currently collect. In facilitating the implementation of PCOMs into routine practice in these instances, we have worked collaboratively with specialist palliative care services and health professionals to develop 'The RESOLVE PCOM Implementation Strategy'. This represents a complex intervention in that it 'comprise(s) (of) multiple interacting components' (see table 4 for a detailed breakdown of each component).[15] The first two of these are general elements that are mostly prespecified and provided to sites in a standardised manner. The next four are tailored closely to the needs of each site. The final component of this implementation strategy is the role of the quality improvement facilitator which permeates both general and site-specific implementation strategies.

**Table 2** An overview of the research design for this study

| Strand | Data collection method | Data collection timepoints | Reason/output |
|---|---|---|---|
| 1 | Routine outcomes | Baseline and follow-up | To map changes in use (ie, implementation status) of person-centred outcome measures over time (Research Question 1) |
| 2 | Survey | Initial piloting phase and then at baseline and follow-up | To determine how changes in person-centred outcome measures over time (RQ1) use relates to changes in participants' coherence, engagement, action and reflexive monitoring of measures (Research Question 2) |
| 3 | Focused interviews | Towards end of project | To determine why? how? context? of any changes (ie, mechanisms) (Research Questions 2&3) |

**Table 3** Description of the core set of PCOMs that we aim to implement into routine palliative care practice as part of the RESOLVE project

| Measure | Description | Frequency/timing of collection |
|---|---|---|
| Palliative Phase of Illness[17] | Palliative Phase of Illness is a measure which describes the urgency of care needs for a person receiving palliative care. It does so by describing four distinct phases of a patient's illness, including: stable, unstable, deteriorating, dying and deceased. These phases are measured through determining the care needs of a patient and/or their family and provide a clinical indication of a patient's Phase of Illness which can be used to inform care planning. | **In-patients (hospice)** ► Daily **Community** ► Each contact |
| Integrated Palliative care Outcomes Scale (IPOS)[18 19] | A holistic, well-validated and global measure of symptom burden that uses 10 questions (scored on a 0–4 Likert-type scale) to assess the most important symptoms and concerns of patients affected by life-limiting illnesses across physical, psychological, social and existential domains of well-being. There are two forms of IPOS; patient-IPOS (where patients complete the questionnaire as a self-report) and staff-IPOS (a proxy version which is completed by staff). | **In-patients (hospice)** ► Initial assessment ► Change in Phase of Illness ► End of episode **Community** ► Each contact |
| Australia-modified Karnofsky Performance Status (AKPS)[20] | Assesses a patient' overall performance/functional status across three dimensions: activity, work and self-care. Healthcare professionals use their observations of patients' ability to perform everyday tasks and scores them at 10% increments between 0% (ie, the patient is dead) and 100% (ie, no complaints or evidence of disease). | |

PCOMs, person-centred outcome measures.

All elements of this implementation strategy are rooted in an assessment of learning needs at each site, the development of rapport with staff and a visible response to ongoing feedback. They are intended to empower clinical staff to appropriately use PCOMs, including guidance on ensuring the frequency/timing of collection is suitable for the setting in which they are working (see table 3). This may take the form of sharing formal knowledge about PCOMs and how to use them, practical and technical knowledge about IT systems and/or experiential knowledge about the use of PCOMs in practice. Where PCOM use is affected by team or organisational issues,

quality improvement facilitators can empower clinical staff to address these issues and may also play an advocacy role through liaison with senior clinical staff or management. A more in-depth version of The RESOLVE PCOM Implementation Strategy (mapped against the different components included in the template for intervention description and replication (TIDIeR) checklist)[21] can be found in online supplemental file 1.

### Theoretical approach and causal assumptions

As part of their guidance on process evaluations, Moore *et al*[15] recommend drawing on theory to propose causal

**Table 4** An outline of the main components of The RESOLVE PCOM Implementation Strategy

| Implementation component | Characteristics |
|---|---|
| *General implementation strategies* | |
| 1. Development of educational resources | Providing healthcare professionals with resources (eg, brief instructional videos and pocket guides/outcomes manual) aiming to enhance their knowledge and skills of what outcome measures are, and how to use the right measures at the right time in clinical practice |
| 2. Workshop and conference events | Developing and running national and regional workshop events that aim to address common challenges to implementation and facilitate collaborative learning networks between palliative care services |
| *Site-specific implementation strategies* | |
| 3. Determining organisational needs | A baseline assessment—developed through initial interview data and additional liaison with service leads at each site—aimed at understanding the site-specific challenges to implementing PCOMs so that the intervention can be tailored to each site's needs. This includes understanding whether sites want help with better implementing a measure that is already used, or with implementing new measure(s) across or in specific settings |
| 4. Formal training | Face-to-face and online educational sessions/presentations with healthcare professionals at sites in order to address local implementation issues/challenges |
| 5. IT support | Providing informational, practical and technical support to ensure that each site has the analytic capacity to input outcomes data into, and extract it back out of, electronic systems |
| 6. Reporting and feedback | Providing sites with tailored, service-level reports and feedback of their outcomes data to motivate continued use |
| *Running through general and site-specific implementation strategies* | |
| 7. Quality improvement facilitator | Working directly with sites by local site liaison, championing the implementation of PCOMs into practice, identifying and responding to local challenges/needs, keeping PCOMs on the agenda and acting as a direct link between the research team and sites. An important aspect of this role is to make judicious use of the implementation strategy elements, rather than necessarily deliver the strategy as a whole |

PCOMs, person-centred outcome measures.

assumptions underpinning how and why The RESOLVE PCOM Implementation Strategy is expected to work. To this end, the components of our implementation strategy are theoretically informed by Normalisation Process Theory (NPT).[22 23] The reason behind selecting NPT is because there is evidence of its effectiveness and value in implementing complex interventions within healthcare settings.[23]

NPT is a type of implementation theory that describes the different types of 'work' that individuals and organisations do in order to 'normalise' complex interventions into clinical practice (ie, such deep embedment that they become 'invisible'). It does so through proposing four inter-related constructs:

### Coherence
The 'sense-making work' that individuals and teams do in order to understand PCOMs.

### Cognitive participation
The 'relational work' that occurs in legitimising PCOMs and building everyday clinical practice around using them.

### Collective action
The 'operational work' that is performed in order to use PCOMs, including the skills-sets of a workforce and the organisational resources in place that enable the use of PCOMs.

### Reflexive monitoring
The 'appraisal work' through which people assess and evaluate the relative value of PCOMs.[22 24]

We therefore propose that The RESOLVE PCOM Implementation Strategy will facilitate the implementation of PCOMs into routine practice in four ways: (1) by enhancing professionals' understanding and awareness of how and why to use PCOMs (ie, improving coherence); (2) by creating an organisational culture where PCOMs are valued and seen as a legitimate way to improve patient care (ie, enhancing cognitive participation); (3) by ensuring that systems and resources are in place that allows healthcare professionals to easily and meaningfully use PCOMs in everyday practice (ie, facilitating collective action); and (4) by providing meaningful feedback on PCOMs data as to motivate people to continue using them (ie, enhancing reflexive monitoring).

## Patient and public involvement
We support the UK Standards for Public Involvement in research as outlined by the National Institute of Health Research.[25] We have already conducted patient public involvement for strand one of this project (the collection and use of routine outcomes data) in which we consulted with a diverse group of patients about implementation. Furthermore, the RESOLVE project has recruited a lay representative who attends regular project meetings and will be involved in the discussions regarding the interpretation and dissemination of study findings throughout the duration of the project.

## Strand one: mapping change through routinely collected outcomes data
### Sampling and recruitment
In recruiting to this study, we will send an invitation letter and participant information sheet to local collaborators at each site. Should sites wish to participate, they will be required to sign and return the study invitation letter. We expect all 11 RESOLVE-affiliated sites to participate in this project.

### Data collection
The RESOLVE team have already been working with sites in implementing the palliative Phase of Illness, IPOS and AKPS into routine clinical practice. Sites will be required to submit pseudonymised data related to these PCOMs to the RESOLVE team at baseline and follow-up (between 10 and 12 months later). A more detailed description of the data specification can be found in online supplemental file 2.

### Data analysis
In order to map whether or not the uptake and routine use of PCOMs changes over time, we will use descriptive and analytical statistics to report the levels of missing data at baseline and follow-up (missing complete measures, and missing items).

For each site at baseline, we will assess the missing outcome data at three timepoints: (a) start of episode of care, (b) first change of Phase of Illness and (c) end of episode of care. To highlight the change in routine use of PCOMs, we will compare the missing outcomes data between baseline and follow-up at these three timepoints. We will assess missing data on the level of these subscales: physical symptoms (10 items), psychological/emotional symptoms (4 items) and communication/practical issues (3 items). We defined missing outcome data in physical symptoms as 4 or more items missing data out of the 10 items at one point of time. For emotional and communication subscales, missing outcome data were defined as missing at least one item in any of them. We will also compare between the population samples at baseline and follow-up to investigate the comparability of both groups. The variables that we will use to do this will be gender, age, primary diagnosis and whether a patient lives alone. We will then use t-tests to compare continuous data (eg, age) with mean and SD, and $\chi^2$ to compare categorical data (eg, gender). All statistical analyses will be conducted at alpha value of 0.05 two-sided level of significance.

## Strand two: online survey data
Conducting survey research will help us to understand whether The RESOLVE PCOM Implementation Strategy has enhanced healthcare professionals' understandings of, skills in using and ability to build organisational practices around, PCOMs. The design of the survey strand of this project is informed by Kelley et al's[26] guidance on

good practice on the conduct and reporting of survey research.

### Sampling and recruitment

We estimate that, when combined, the number of staff across all of the participating services equates to approximately 550 participants. Whilst we aim to distribute this survey to all of these potential participants, we recognise the challenges to online surveys (eg, participants interests/attitudes to research, communication, structure and length of survey, etc.).[27] Therefore, of this number, due to the positive relationships that we have fostered with sites throughout this project, we expect between a 50% and 60% response rate. Thus, for the survey part of this research, we estimate between 275 and 330 participants.

We will use a purposive maximum variation sampling technique[28] as a guide for recruitment, and sample across the following purposive criteria: role, seniority, experience and settings of care (in-patient, outpatient/day therapy and home-based/community). Through recruiting in this way, we aim to sample an accurate cross-section of the different settings in which PCOMs are used, and the different types of staff members who use them within these settings. This will translate into recruiting staff at different levels/seniority (eg, those at a medical, senior staff and junior level) and of differing roles (eg, doctors, nurses, healthcare assistants, educators, allied healthcare professionals).

To recruit participants, local collaborators at each site will be asked to disseminate an online survey link to healthcare professionals within their organisations. The introductory part of the survey will include the relevant participant information that explains what the purpose of the survey is, why they have been selected, and how data will be used. It will also state that by completing and submitting the survey that they are providing consent to participate in this study. To optimise recruitment and survey completion, we will offer participating sites the incentive of a textbook and/or journal subscription for the organisation for 1 year.

### Data collection

In keeping with the theoretical framework of this project, we will collect survey data using an adapted version of the NPT survey (The Normalisation MeAsure Development questionnaire (NoMAD). This is a 23-item structured questionnaire that has been developed by the authors of NPT and measures the dynamic processes that underpin the implementation of PCOMs into clinical practice from the perspectives of healthcare professionals directly involved in their use. The NoMAD measure has been found to be an acceptable and psychometrically strong measure which reports good face validity, construct validity and internal consistency.[29]

Our version of the NoMAD survey consists of three parts which comprise of a combination of closed-ended and open-ended questions. Part A asks participants to provide demographic information about themselves. Part B asks short questions on what PCOMs participants use and how they use them in everyday practice. Part C is split into four sections, each of which corresponds and asks questions in relation to the different constructs of NPT (coherence, cognitive participation, collective action and reflexive monitoring). This survey can be seen in more detail in online supplemental file 3.

We will host our survey online using RedCap software in which participants will be sent a link through which they are able to access and complete the survey. The acceptability of this method, alongside the intelligibility of survey questions, will be piloted on a select number of participants to ensure that they understand the questions being asked and are easily able to complete the online form. Following this, we will make any necessary minor amendments to the NoMAD survey and then administer this to participants at baseline and then at follow-up (10–12 months after baseline collection).

### Data analysis

We will conduct four analyses of survey data:

(i) Response rate by capturing the number of participants who completed the NoMAD survey compared with the proportion of those who were approached. (ii) Descriptive statistics to describe the general characteristics of healthcare staff who participated in the survey. (iii) Comparison between respondent answers at baseline and follow-up using statistics tests such as paired t-test and McNemar test (when needed) to indicate any changes in PCOMs usage and understanding. All statistical analyses will be conducted at alpha value of 0.05 two-sided level of significance. We recognise that there are numerous potential limitations/challenges of conducting follow-up surveys 12 months after baseline. These include losing participants to follow-up, withdrawal, staff turnover, changed positions, each of which may lead to decrease the sample size at follow-up time. Thus, we decided if any group lost more than 70% of its participants, we will recruit other health staff from the site (who did not participate at baseline) to complete our follow-up sample. Therefore, we will change the analysis plan and treat both groups two independent samples. (iv) Free-text survey responses will be deductively thematically analysed using NPT constructs (coherence, cognitive participation, collective action and reflexive monitoring) as a framework, while also allowing for inductive thematic analysis if responses do not fit within the framework.

### Strand three: focused interview data

For strand three, we will collect focused interview data. The collection of qualitative interview data will be able to provide further insight into research questions two and three by providing rich and insightful data on the processes, mechanisms and contextual factors that underpin the ways in which The RESOLVE PCOM Implementation Strategy does, or does not, work to implement PCOMs into routine practice.[30]

## Sampling and recruitment

For each site, depending on organisational size, we will recruit between two and five participants. To do this, we will use purposive sampling[28] (based on roles, disciplines and experience) as a guide to recruit staff who are directly involved in the implementation of PCOMs across different settings, and able to offer a distinct insight/experience into the issues identified from the survey about each of the NPT constructs. Recruitment will be achieved through working with service leads at each site to approach the relevant participants for interview in this study. Participants will then be approached by their service leads with a participant information sheet. Willing participants will then be required to provide informed consent prior to participation.

### Data collection

We will conduct a single-focused interview with each participant towards the end of the project (ie, when we finish our implementation work). These interviews will be used as an opportunity to generate rich insight and details of the processes, mechanisms and contextual factors that underpin how The RESOLVE PCOM Implementation Strategy does, or does not, work to facilitate the implementation of PCOMs into routine palliative care practice. The interview guide (see online supplemental file 4) will consist of a series of questions that relate to the different components of NPT (coherence, cognitive participation, collective action and reflexive monitoring) and will present participants with an opportunity to reflect on how their use of PCOMs has changed over time, and how The RESOLVE PCOM Implementation Strategy has impacted this. In mitigating for the COVID-19 pandemic,

these interviews will take place online (on whichever platform is most convenient for services) at a time that works for them.

### Data analysis

Data will be analysed in NVivo using a thematic framework approach.[31] NPT constructs (coherence, cognitive participation, collective action and reflexive monitoring) will provide a theoretical framework that will be applied deductively to interview transcripts, while also allowing for inductive thematic analysis of responses and outliers that do not fit within the framework. This will be through conducting the following interconnected steps: (1) transcription; (2) familiarisation; (3) coding; (4) developing an analytic framework based on NPT; (5) indexing this framework to transcripts; (6) charting data into a framework matrix (and then performing between-case and within-case analyses for each theme) and (7) interpreting the data by using theory.

### Outputs: integration of data sets

Each strand of this project will produce different outputs. In strand one, we will use routine outcomes data to produce a traffic light system in which green represents that PCOMs are used appropriately (eg, in line with the timings/frequencies of collections outlined in table 3), amber that PCOMs are used but inconsistently (eg, they are used but not in line with the timings/frequencies of collection in table 3) and red that PCOMs are not being used at all. At a glance, this will provide an objective indicator of whether there has been a change (from baseline to follow-up) in the uptake and use of each PCOM across settings, thus enabling us to determine whether The

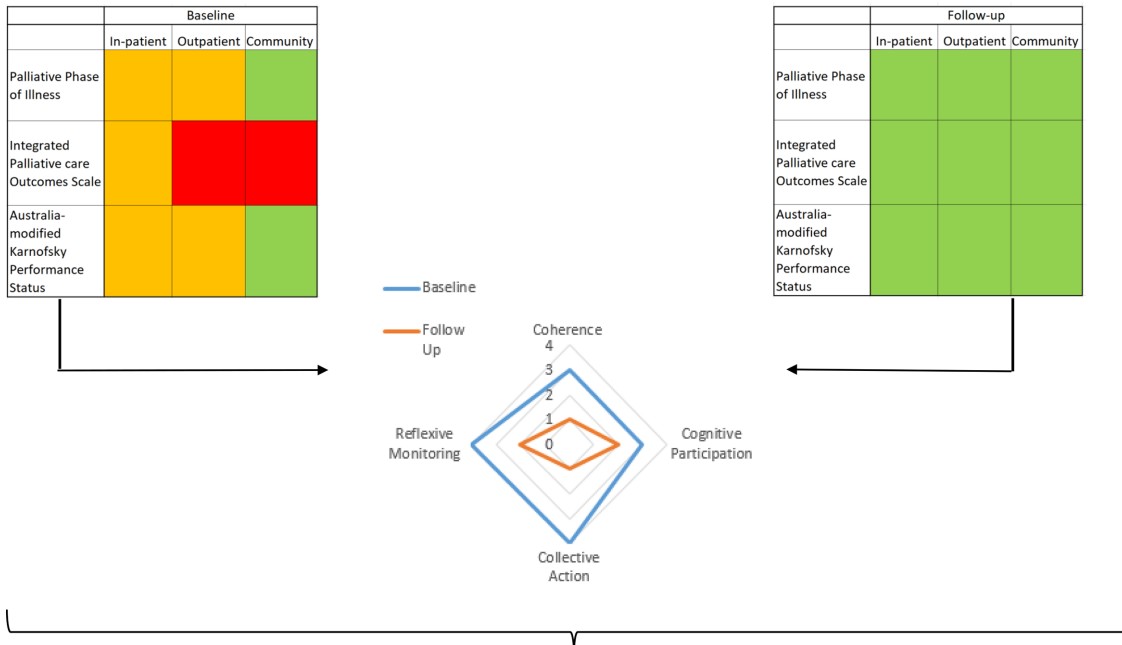

Qualitative data explaining processes, mechanisms, and contextual factors of how our complex intervention led to change.

**Figure 1**  Example of how outputs will be integrated after analysis.

RESOLVE Implementation Strategy has been successful in implementing PCOMs into routine palliative care practice.

For strand two, we will produce outputs for NoMAD survey data in the form of spider plot graphs. By calculating how mean changes in each NPT component changes from baseline to follow-up, this data will be able to complement our analyses of routine outcomes data by seeing if changes in use of PCOMs status coincides with changes in components of NPT.

For strand three, data from interviews will complement routine outcomes and NoMAD survey data by illuminating the mechanisms as to how and why the use of PCOMs have (or have not) changed over time. It will also be able to provide important information about the contextual/organisational factors that impacted this process.

Figure 1 demonstrates how the integration of outputs produced from the analysis of quantitative and qualitative data may look. It will provide a snapshot that will be able to demonstrate the effectiveness The RESOLVE Implementation Strategy at facilitating the implementation of PCOMs into routine palliative care practice. It will also provide us with explanations as to how, why and in what contexts our intervention works/needs refining so that it can be tailored for particular organisational contexts and/or settings of care.

## Ethics and dissemination

Ethical approval for this project was obtained on 16 September 2020 from the East of England—Cambridge Central NHS Research Ethics Committee (ref: 20/EE/0188). Ethical approval for strands 2 and 3 of this study was obtained on 18 January 2021 from Hull York Medical School ethics committee (2105).

We will publish the findings of this study in peer-reviewed journals and abstracts will be submitted to national and international conferences. We will also disseminate this work to patients, healthcare professionals, academics and anybody else who is interested in this work through various non-traditional routes. These will include the following:

► Producing lay briefs of research findings that we will disseminate to patients and service users through our Palliative Daycare Patient Group and the Patient and Public Involvement and Engagement (PPIE) network at the University of Hull.
► Producing policy briefs of research findings for policy-makers and commissioners.
► Sharing the results of research and study newsletters on the RESOLVE and Wolfson Palliative Care Research Centre websites and sharing them on social media.
► Delivering educational workshops to clinicians in order to feedback site-specific findings on what does and does not work with regard to implementing PCOMs within their service.
► Use the Hospice UK's ECHO Network for Outcomes—monthly national interactive webinars to disseminate

the results to clinicians, allied healthcare professionals and others who have an interest in palliative care.

**Contributors** Conceptualisation and design—AB, AMK, MP and FEMM. Development of analysis plan—AB, AMK and FEMM. Leading the writing process and drafting the original article—AB. Critically reviewed article for important content—AB, MP, MS, AMK, KS, JB, MJ and FEMM. All authors take responsibility for the protocol and approved the final version of this paper.

**Funding** This work was supported by Yorkshire Cancer Research, grant number (L412). Professor Fliss Murtagh is a National Institute for Health Research (NIHR) Senior Investigator. The views expressed in this article are those of the author(s) and not necessarily those of the NIHR, or the UK Department of Health and Social Care.

**Competing interests** The author(s) declared no potential conflicts of interest with respect to the research, authorship and/or publication of this article.

**Patient consent for publication** Not required.

**Provenance and peer review** Not commissioned; externally peer reviewed.

**ORCID iDs**
Andy Bradshaw http://orcid.org/0000-0003-1717-1546
Jason Boland http://orcid.org/0000-0001-5272-3057

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
