## [Reviewer comments · BMJ Open]

ARTICLE DETAILS

TITLE (PROVISIONAL)	Implementing Person-Centred Outcome Measures (PCOMs) into routine palliative care: Protocol for a mixed-methods process evaluation of the RESOLVE PCOM Implementation Strategy
AUTHORS	Bradshaw, Andy; Santarelli, Martina; Khamis, Assem M; Sartain, Kathryn; Johnson, Miriam; Boland, Jason; Pearson, Mark; Murtagh, Fliss

VERSION 1 – REVIEW

REVIEWER	Bolzani, Anna Klinikum der Universitat Munchen, Department of Palliative Medicine
REVIEW RETURNED	14-May-2021

GENERAL COMMENTS	Thank you for the opportunity to review the manuscript “Implementing Person-Centred Outcome Measures (PCOMs) into routine palliative care: Protocol for a mixed-methods process evaluation of the RESOLVE PCOM Implementation Strategy”. The research is highly relevant and will be helpful for patients, health care professionals and researchers. However, I have some open questions and comments about this study protocol that the authors should consider. Abstract • Please describe which routine outcomes data will be collected (e.g. mentioning the measures used).• Please include after how many months data will be collected (for Strand one and two). Methods and analysis Strand one: Mapping change through routinely collected outcomes data • Data analysis, p14, line 22f: it seems like the ‘subscales’ are the ones commonly used for IPOS. If so, please clarify that.• If the patient version of the IPOS will be used: Patients sometimes prefer not to answer specific items due to a variety of reasons. Will this be defined as ‘missing data’ as well? Please specify this in more detail.• Please define all variables used to analyse ‘the comparability of both groups’ (p14, l34).• I would assume that subgroup analyses in this project are of high interest (for example missing outcome data for subgroups: oncology vs non-oncology patients; different settings; different size of participating study locations, time between start and end of period of care). However, I have not found any information about it in this section. If you plan to conduct any subgroup analyses in
--

	Strand one, please add this information and the according analysis plan.  • It is unclear to me, if the study sites receive specific instructions on how often PCOM should be used (every day, every time a change of Phase of Illness occurs, site-specific definitions). It would be great if any specific guidelines for each PCOM could be added to table 3 (for example). • Following the comment before, I am also wondering if there are any inclusion/ exclusion criteria for the patients. For example, what happens if a patient has a change of Phase of Illness within the first day (therefore, the date of a and b might be the same)? Are there any planned subgroup analyses around time periods between change of Phase of Illness? What happens if start and end of episode of care are within a short time period (for example <24 h)? • For the sake of completeness: could you please add details to your planned analyses, that is alpha value, one-sided/two-sided, etc. Strand two: Online survey data  • Could you please clarify if you will send out the survey to all approximately 550 participants or if you use your defined sampling technique to select a smaller number of participants that will be recruited through the local collaborators? If you select a smaller number of participants: what is your planned sample size? • As mentioned in Strand one, please add more information about the statistical analysis including alpha values, etc. Outputs: integration of datasets  • You describe the traffic light system to categorise your data collected during Strand one. Could you please add the thresholds used to define appropriate use, inconsistent use and not being used at all? • My apologies for the repeated comment, but I am again wondering if you are planning any subgroup analyses at that stage? (Especially different settings, different patient groups, etc.) If so, please specify these analyses. General comments:  • All tables/figures: please explain abbreviations. • From my understanding, the participating sites already use PCOM in routine care. How do you ensure that the RESOLVE PCOM Implementation Strategy is (or is not) relevant for sites without PCOM use in the first place? • “Variable Views on Care” (document Data item specification; item 36) No operationalisation and analysis plan of this variable is described in the main text. Please add if this variable will inform the analysis of any Strand.
--	--

REVIEWER	Ahlström, Gerd Lunds Universitet, Health Sciences
REVIEW RETURNED	15-May-2021

GENERAL COMMENTS	Thanks for giving me the opportunity to read this article. The study treats a research area highly relevant for BMJ OPEN. Generally, this evaluating of successful implementation strategies is an important issue for the palliative care service in every country. However, I notice that the author already finished the data collection 2019 according to the recently published paper with the title “Implementing person-centred outcome measures in palliative care:
--

	An exploratory qualitative study using Normalisation Process Theory to understand processes and context” in Palliative Medicine 2021; https://journals.sagepub.com/doi/epub/10.1177/0269216320972049 According to the description in the protocol manuscript, are the published interview data answering research question 3 and corresponding to the last strand of the project (page 17-19). The guideline for BMJ OPEN is that a protocol paper should report planned or ongoing studies and project in the final step is outside the journal scope. Also, the dates for the study should be written in the manuscript.
--	--

VERSION 1 – AUTHOR RESPONSE

Reviewer 1	
Comment 1: Thank you for the opportunity to review the manuscript “Implementing Person-Centred Outcome Measures (PCOMs) into routine palliative care: Protocol for a mixed-methods process evaluation of the RESOLVE PCOM Implementation Strategy”. The research is highly relevant and will be helpful for patients, health care professionals and researchers. However, I have some open questions and comments about this study protocol that the authors should consider.	Response 1: Thank you for taking the time to review this paper. We have responded to all comments and questions below.
Abstract Comment 2: Please describe which routine outcomes data will be collected (e.g. mentioning the measures used).	Response 2: We have added these in (see red text on p.2, lines 9-10). It now reads: ‘Methods and analysis: Multistrand, mixed methods process evaluation. Strand one will collect routine outcomes data (palliative Phase of Illness, Integrated Palliative care Outcomes Scale, Australia-modified Karnofsky Performance Status) to map the changes in use of outcome measures over 12 months.’
Comment 3: Please include after how many months data will be collected (for Strand one and two)	Response 3: We have included this information in the abstract (see red text on p.2, lines 9-12). It now reads: ‘Methods and analysis: Multistrand, mixed methods process evaluation. Strand one will collect routine outcomes data (palliative Phase of Illness, Integrated Palliative care Outcomes Scale, Australia-

	modified Karnofsky Performance Status) to map the changes in use of outcome measures over 12 months (July 2021-July2022). Strand two will collect survey data over the same 12-month period to explore how professionals' understanding of, skills in using, and ability to build organisational practices around, outcome measures change over time.'
Methods and analysis	
Strand one: Mapping change through routinely collected outcomes data Data analysis, p14, line 22f: it seems like the 'subscales' are the ones commonly used for IPOS. If so, please clarify that:	
Comment 4: If the patient version of the IPOS will be used: Patients sometimes prefer not to answer specific items due to a variety of reasons. Will this be defined as 'missing data' as well? Please specify this in more detail.	Response 4: The focus of this paper is on understanding staff perspectives of the RESOLVE implementation strategy and how it impacts their ability to collect outcomes data. The details of the type of measure (patient- or proxy-collected) will be reported elsewhere. Whilst we will report on overall levels of missing data, the reasons for missingness are not part of this work but will be undertaken in a separate study on missing outcomes data. As such, we cannot report on this in the proposed study.
Comment 5: Please define all variables used to analyse 'the comparability of both groups' (p14, l34)	Response 5: We have added in the variables that we will use to analyse the comparability of both groups. (see red text on page 15, lines 14-16. It now reads: 'We will also compare between the population samples at baseline and follow-up to investigate the comparability of both groups. The variables that we will use to do this will be gender, age, primary diagnosis, and

	whether a patient lives alone. We will then use t-tests to compare continuous data (e.g., age) with mean and SD and chi-square to compare categorical data (e.g., gender).'
Comment 6: I would assume that subgroup analyses in this project are of high interest (for example missing outcome data for subgroups: oncology vs non-oncology patients; different settings; different size of participating study locations, time between start and end of period of care). However, I have not found any information about it in this section. If you plan to conduct any subgroup analyses in Strand one, please add this information and the according analysis plan.	Response 6: We do not plan to do subgroup analyses for missing data for the purposes of strand 1. We are only focusing on whether outcomes data were collected or not. We will do subgroup analyses on missing outcomes data and different settings in a separate piece of work.
Comment 7: It is unclear to me, if the study sites receive specific instructions on how often PCOM should be used (every day, every time a change of Phase of Illness occurs, site-specific definitions). It would be great if any specific guidelines for each PCOM could be added to table 3 (for example).	Response 7: Study sites will be given specific instructions on how PCOMs should be used. We have updated the manuscript in two places to reflect this. Firstly, we have added an extra column to Table 3 (on page 10) titled 'Frequency/timing of collection', outlining when these measures should be collected in inpatient and community settings. We have also updated the description of our intervention to include explicit reference to this (see red text on p.11, lines 17-19). It now reads: 'They are intended to empower clinical staff to appropriately use PCOMs, including guidance on the frequency/timing of collection is suitable for the setting in which they are work (see table 3).'
Comment 8: Following the comment before, I am also wondering if there are any inclusion/ exclusion criteria for the patients. For example, what happens if a patient has a change of Phase of Illness within the first day (therefore, the date of a and b might be the same)? Are there any planned subgroup analyses around time periods between change of Phase of Illness? What happens if start and end of episode of care are	Response 8: The palliative Phase of Illness measure is designed to be measured daily and not more frequently; we would not therefore expect change of Phase within the first day. We are not planning

within a short time period (for example <24 h)?	any subgroup analysis around the time periods for this piece of work.
Comment 9: For the sake of completeness: could you please add details to your planned analyses, that is alpha value, one-sided/two-sided, etc.	Response 9: This is now included in the manuscript (see red text on page 15, lines 17-18). It now reads: 'The variables that we will use to do this will be gender, age, primary diagnosis, and whether patient lives alone. We will then use t-tests to compare continuous data (e.g., age) with mean and SD and chi-square to compare categorical data (e.g., gender). All statistical analyses will be conducted at alpha value of 0.05 two-sided level of significance.'
Strand two: Online survey data	
Comment 10: Could you please clarify if you will send out the survey to all approximately 550 participants or if you use your defined sampling technique to select a smaller number of participants that will be recruited through the local collaborators? If you select a smaller number of participants: what is your planned sample size?	Comment 10: We do aim to send this survey out to as many of the estimated 550 participants as possible. We have made this clearer in the manuscript (see in red on page 16, lines 3-6). It now reads: 'We estimate that, when combined, the number of staff across all of the participating services equates to approximately 550 participants. Whilst we aim to distribute this survey to all of these potential participants, we recognise the challenges to online surveys (e.g., participants interests/attitudes to research, communication, structure and length of survey etc.).²⁷ Therefore, of this number, due to the positive relationships that we have fostered with sites throughout this project, we expect between a 50%-60% response rate.'
Comment 11: As mentioned in Strand one, please add more information about the statistical analysis including alpha values, etc.	Comment 11: This is now included in the manuscript (see red text on page 18, lines 8-9). It now reads: 'Comparison between respondent answers at baseline and follow-up

	using statistics tests such as paired t-test and McNemar test (when needed) to indicate any changes in PCOMs usage and understanding. All statistical analyses will be conducted at alpha value of 0.05 two-sided level of significance.'
Outputs: integration of datasets	
Comment 12: You describe the traffic light system to categorise your data collected during Strand one. Could you please add the thresholds used to define appropriate use, inconsistent use and not being used at all?	Response 12: We have added this information in. See in red on page 20, lines 17-19). It now reads: 'Each strand of this project will produce different outputs. In strand one, we will use routine outcomes data to produce a traffic light system in which green represents that PCOMs are used appropriately (e.g., in line with the timings/frequencies of collections outlined in table 3), amber that PCOMs are used but inconsistently (e.g., they are used but not in line with the timings/frequencies of collection in table 3), and red that PCOMs are not being used at all.'
Comment 13: My apologies for the repeated comment, but I am again wondering if you are planning any subgroup analyses at that stage? (Especially different settings, different patient groups, etc.) If so, please specify these analyses.	Response 13: As per the above comments, we are not planning any other subgroup analyses for this piece of work
General comments:	
Comment 14: All tables/figures: please explain abbreviations.	Response 14: All abbreviations have been explained in the tables and figures. See changes in red.
Comment 15: From my understanding, the participating sites already use PCOM in routine care. How do you ensure that the RESOLVE PCOM Implementation Strategy is (or is not) relevant for sites without PCOM use in the first place?	Response 15: An important process in ensuring that the RESOLVE PCOM implementation strategy is relevant for, and can be tailored to the needs of, sites (both who use and do not use certain measures) is implementation component 3 (in table 4): 'determining organisational needs'. In making clearer how we will ensure that our strategy is relevant in both instances, we have made

	changes to two parts of the manuscript. Firstly, on page 11 (lines 3-7) we have made the following change: ‘Whilst all participating sites collect PCOMs in some form, their use is variable. For example, some use all three measures but not consistently across settings, others may use certain measures consistently in some settings but not others, whilst some are seeking to introduce new measures that they do not currently collect. In facilitating the implementation of PCOMs into routine practice in these instances, we have worked collaboratively with specialist palliative care services and health professionals to develop ‘The RESOLVE PCOM Implementation Strategy’.’ We have also expanded on the description of intervention component 3 in Table 4 (see in red on page 12). It now reads: ‘A baseline assessment - developed through initial interview data and additional liaison with service leads at each site – aimed at understanding the site-specific challenges to implementing PCOMs so that the intervention can be tailored to each site’s needs. This includes understanding whether sites want help with better implementing a measure that is already used, or with implementing new measure(s) across or in specific settings’
Comment 16: “Variable Views on Care” (document Data item specification; item 36) No operationalisation and analysis plan of this variable is described in the main text. Please add if this variable will inform the analysis of any Strand.	Response 16: This data item is being collected as part of the same dataset that we gained ethical approval for, but for other work. Therefore, it will not be

	analysed in this study.
Reviewer 2	
Comment 1: Thanks for giving me the opportunity to read this article. The study treats a research area highly relevant for BMJ OPEN. Generally, this evaluating of successful implementation strategies is an important issue for the palliative care service in every country. However, I notice that the author already finished the data collection 2019 according to the recently published paper with the title “Implementing person-centred outcome measures in palliative care: An exploratory qualitative study using Normalisation Process Theory to understand processes and context” in Palliative Medicine 2021; https://journals.sagepub.com/doi/epub/10.1177/0269216320972049 According to the description in the protocol manuscript, are the published interview data answering research question 3 and corresponding to the last strand of the project (page 17-19). The guideline for BMJ OPEN is that a protocol paper should report planned or ongoing studies and project in the final step is outside the journal scope.	Response 1: Thank you for taking the time to review this paper. The published interview data from the paper that you reference does not answer research question 3 of this protocol paper. The already-published paper in Palliative Medicine aimed to understand and explain the causal mechanisms that underpin the successful implementation of outcome measures into routine practice. Whilst these data informed part of this protocol (i.e., understanding challenges to implementation in order to develop parts of the RESOLVE PCOM implementation strategy), it is separate to the study being proposed within this protocol. Rather than looking at factors impacting the implementing outcome measures per se, the interviews within this protocol are specifically designed to understand the mechanisms through which the RESOLVE PCOM implementation strategy does (or does not) work to implement outcome measures in practice (see p.18, lines 21-24). This is to learn implementation lessons regarding the strategies that we may use to implement complex interventions (in this case PCOMs) into healthcare settings (in this case palliative care). In being more explicit about the differences in these two pieces of work, we have expanded on the ‘The RESOLVE project’ section (see in red on page 6, lines 1-4): ‘As part of this study, we conducted an exploratory

	qualitative study in which semi-structured interviews were used to understand the processes and causal mechanisms that underpin the successful implementation of PCOMs into routine practice. ¹³ Building on insights from this study about implementation and our collaborative work with sites, this study protocol describes a process evaluation of the complex intervention – named ‘The RESOLVE PCOM Implementation Strategy’ – aimed at facilitating the integration of PCOMs into routine practice.
Also, the dates for the study should be written in the manuscript.	We have added the expected time period of this work in the abstract (see red text on p.2, lines 9-12). It now reads: ‘Methods and analysis: Multistrand, mixed methods process evaluation. Strand one will collect routine outcomes data (palliative Phase of Illness, Integrated Palliative care Outcomes Scale, Australia-modified Karnofsky Performance Status) to map the changes in use of outcome measures over 12 months (July 2021-July 2022). Strand two will collect survey data over the same 12-month period to explore how professionals’ understanding of, skills in using, and ability to build organisational practices around, outcome measures change over time.’